# TTSDS2: Resources and Benchmark for Evaluating Human-Quality Text to Speech Systems

**Christoph Minixhofer, Ondrej Klejch, Peter Bell**
Centre for Speech Technology Research
University of Edinburgh
`{christoph.minixhofer,o.klejch,peter.bell}@ed.ac.uk`

## Abstract

Evaluation of Text to Speech (TTS) systems is challenging and resource-intensive. Subjective metrics such as Mean Opinion Score (MOS) are not easily comparable between works. Objective metrics are frequently used, but rarely validated against subjective ones. Both kinds of metrics are challenged by recent TTS systems capable of producing synthetic speech indistinguishable from real speech. In this work, we introduce Text to Speech Distribution Score 2 (TTSDS2), a more robust and improved version of TTSDS. Across a range of domains and languages, it is the only one out of 16 compared metrics to correlate with a Spearman correlation above 0.50 for every domain and subjective score evaluated. We also release a range of resources for evaluating synthetic speech close to real speech: A dataset with over 11,000 subjective opinion score ratings; a pipeline for recreating a multilingual test dataset to avoid data leakage; and a benchmark for TTS in 14 languages.

## 1 Introduction

### 1.1 Background and impact of TTS evaluation

Text to Speech models have significantly advanced recently, achieving a level of quality where synthetic speech can be indistinguishable from real speech (Eskimez et al., 2024). Since subjective evaluation of TTS using listening tests is difficult and resource-intensive, some recent works have partially or fully replaced subjective metrics with objective ones (Cooper et al., 2024). Recent TTS systems report human-level quality, with listeners sometimes not able to distinguish between real and synthetic speech (Chen et al., 2024; Wang et al., 2024; Li et al., 2023; Shen et al., 2024).

The domains of TTS have shifted as well. Read audiobook speech (Zen et al., 2019; Pratap et al., 2020) used to be the standard training data for TTS, but many modern systems now train on scraped datasets instead. For example, the Emilia dataset contains "diverse and spontaneous speaking styles, including breathing, pausing, repetitions, changes in speed, and varying emotions" (He et al., 2024). Multilingual TTS has seen advances too (Liao et al., 2024), but at the time of writing, no public TTS benchmarks for more than a single language exist. The jump in the quality of recent TTS makes it difficult to say if objective metrics continue to reliably predict continually updated human ratings.

However, the speed of progress for synthetic speech generation outpaces most evaluation efforts: in this work we provide the first public evaluation of 20 systems published between 2022-2024 which controls for speaker identities and dataset domains. We are also the first to provide this kind of comparison beyond English, with a total number of 14 languages, with our pipeline extensible to cover more languages in the future. Providing this comparison between systems can improve efficiency when building new systems or extending existing ones. TTSDS2 also gives us information as to how close synthetic speech is to real speech with state-of-the-art models. Our work's impact on advancements in TTS could be used for positive applications such as improving synthetic voices for people who are at risk of losing the ability to speak through illness. Research in this field can also increase risks, such as identity theft by use of synthetic speech – however, we believe that good evaluation practices can also help assess these risks accurately by providing information about the generative capabilities of current and future systems.

## 1.2 CURRENT STATE OF TTS EVALUATION AND SELECTED SYSTEMS

Subjective ratings collected through listening tests are standard practice in TTS evaluation, the most commonly-used rating being Mean-Opinion-Score (MOS). However, pairwise comparison tests, in which listeners rate the preference of one sample over another on a numeric scale, are gaining popularity, as they can lead to significant results with fewer listeners (Cooper et al., 2024). A popular example of these A/B metrics is Comparative MOS (CMOS). A preference test is also frequently conducted to assess systems' ability to replicate a specific speaker, and is called Speaker Similarity MOS (SMOS). Recent systems often achieve MOS or CMOS scores close to or surpassing real speech, making evaluation more challenging (Wang et al., 2024; Li et al., 2023; Lyth & King, 2024). This has lead to a wide range of objective metrics used in recent works, which we describe in more detail in Section 1.4. Additionally, any variant of subjective evaluation is not comparable between works, as the listeners and surveys differ substantially – however, many TTS systems have released their code and weights, making evaluations like ours possible.

In this paper we select 20 open-source, open-weight TTS systems for evaluation, released between 2022 and 2024, and covering 14 languages. See Appendix A for more details about these systems. It should be noted that we only compare voice-cloning TTS systems which use a speaker reference and transcript to control the output. All but two of our selected systems reported subjective evaluation metrics; however, due to different listeners and datasets, these are not comparable between systems. Objective evaluation is frequent as well, especially for ablation experiments and architecture variants (Casanova et al., 2024). We also find that three systems in this set reported parity with human evaluation, which we define as MOS or CMOS within 0.05 of the ground truth values. Four systems surpassed ground truth scores, meaning listeners *preferred the synthetic to the real speech recordings*, with respect to the questions outlined in these tests.

## 1.3 TTSDS2: DISTRIBUTIONAL, ROBUST AND MULTILINGUAL

The Text-to-Speech Distribution Score (TTSDS) was introduced by Minixhofer et al. (2024) as an objective metric using perceptual factors (such as Speaker Identity, Intelligibility, Prosody) and scores each system based on how close several distributions representing each perceptual factor are to a real data reference in comparison to noise. The average of these scores was shown to correlate with subjective ratings.

In this work, we extend and validate TTSDS using 20 TTS systems, all published in 2022 or later. We increase the robustness of TTSDS scores across 4 differing domains, and increase robustness across languages, leading to the updated **TTSDS2**. We compare correlation to subjective listening test results with 16 open-source metrics and find only TTSDS correlates with $\rho > 0.5$ in all cases, with an average of 0.67. We additionally publish a pipeline which continually recreates a multilingual YouTube dataset and synthesises the samples to provide an up-to-date, uncontaminated and automated ranking of TTS systems across 14 languages – we ensure this pipeline can be easily extended as more languages are covered by modern TTS systems.

## 1.4 OTHER OBJECTIVE METRICS

Objective evaluation of synthetic speech can be grouped into four broad families. In addition to these families we also distinguish *intrusive* and *non-intrusive* metrics. Intrusive metrics require some ground truth speech of the same speaker as a reference. Non-intrusive metrics are reference-free. When the reference does not need to contain the same lexical content, it is described as *non-matching*.

**Signal-based reference metrics**: The oldest group consists of intrusive metrics that compare each synthetic utterance to a matching reference. *Perceptual Evaluation of Speech Quality* (PESQ) (ITU-T, 2001) and *Short-Time Objective Intelligibility* (STOI) (Taal et al., 2011) and *Mel-Cepstral Distortion* (MCD) are the best–known representatives. They were designed for telephone or enhancement scenarios rather than TTS, and require access to the ground-truth waveform.

**MOS-prediction networks**: To predict scores directly, researchers train neural networks that map a single audio signal to an estimated MOS. *MOSNet* (Lo et al., 2019) introduced the idea, and was followed by *UTMOS* (Saeki et al., 2022), its SSL-based successor *UTMOSv2* (Baba et al., 2024), and *NISQA-MOS* (Mittag et al., 2021). *SQUIM-MOS* (Kumar et al., 2023) additionally grounds its prediction by requiring a non-matching reference of the ground truth speech. These methods

Table 1: Feature set used for TTSDS compared to TTSDS2.

| Factor | Features (TTSDS) | Features (TTSDS2) |
|---|---|---|
| GENERIC | Hubert [Hsu et al. (2021)] 
 wav2vec 2.0 [Baevski et al. (2020)] | WavLM [Chen et al. (2022)] activations (actv.) 
 HuBERT [Hsu et al. (2021)] (base) actv. 
 wav2vec 2.0 [Baevski et al. (2020)] (base) actv. |
| ENVIRONMENT | VoiceFixer+PESQ [Liu et al. (2021); ITU-T (2001)] 
 WADA SNR [Kim & Stern (2008)] | (Factor removed) |
| SPEAKER | d-Vector [Wan et al. (2018)] 
 WeSpeaker [Wang et al. (2023b)] | d-Vector [Wan et al. (2018)] 
 WeSpeaker [Wang et al. (2023b)] |
| PROSODY | HuBERT [Hsu et al. (2021)] token length 
 WORLD F0 [Morise et al. (2016)] 
 Prosody embeddings [Wallbridge et al. (2025)] | WORLD F0 [Morise et al. (2016)] 
 HuBERT [Hsu et al. (2021)] speaking-rate 
 Allosaurus [Li et al. (2020)] speaking-rate 
 Prosody embeddings [Wallbridge et al. (2025)] |
| INTELL. | wav2vec 2.0 [Baevski et al. (2020)] WER 
 Whisper [Radford et al. (2023)] (small) WER | wav2vec 2.0 [Baevski et al. (2020)] ASR actv. 
 whisper [Radford et al. (2023)] (small) ASR actv. |

report in-domain correlations; however, recent VoiceMOS challenges (Huang et al., 2024) show that correlation with subjective ratings drops out-of-domain.

**Distributional metrics**: Inspired by the image domain's Fréchet Inception Distance (FID) (Heusel et al., 2017), audio researchers proposed measuring entire corpora rather than single files. *Fréchet Audio Distance* (FAD) (Kilgour et al., 2019) compares embeddings and has since been adapted for TTS (Shi et al., 2024). Distributional metrics require a set of references which do not need to correspond to the synthetic data. The authors of these metrics state the need for thousands of samples, which may be why they have not found more widespread adoption.

**Multi-dimensional perceptual metrics**: Recent work argues that no single score can capture everything listeners care about. *Audiobox Aesthetics* predicts Production Quality, Complexity, Enjoyment, and Usefulness scores for audio (Tjandra et al., 2025).

**Other metrics** Often reported are also *Word Error Rate* (WER) and *Character Error Rate* (CER), computed on Automatic Speech Recognition (ASR) transcripts, as well as *Speaker Similarity* computed as the cosine similarity between speaker representations of synthetic speech and a non-matching reference.

## 2 TTSDS2

The task of synthetic speech generation is inherently one-to-many, meaning there is no single ground truth for any given text. In the following, we denote synthetic variables using ~ to avoid overly relying on sub- or superscripts. We therefore frame its evaluation as a problem of distributional similarity. Let $S$ denote a speech signal and $\mathcal{R}$ be a transformation function that extracts a specific feature representation $\mathcal{R}(S)$, such as one of the features shown in Table 1. Our objective is to quantify how closely the empirical distribution of features from a synthetic dataset, $\tilde{P}(\mathcal{R}(\tilde{S})|\tilde{D})$, matches that of a real dataset, $P(\mathcal{R}(S)|D)$, while remaining distinct from various noise distributions, $P^{\text{NOISE}}(\mathcal{R}(\tilde{S})|D^{\text{NOISE}})$, where $D^{\text{NOISE}}$ is drawn from $\mathfrak{D}^{\text{NOISE}}$ which contains uniform noise, normally distributed noise, all ones and all zeros[*]. We choose noise instead of alternate anchors since it is agnostic to domain, speaker and other confounding factors. An example distribution for the one-dimensional pitch feature can be seen in Figure 1.

To achieve this, TTSDS2 employs a factorised evaluation framework, assessing distributional similarity across the following perceptually-motivated aspects of speech: (i) GENERIC: Overall distributional similarity, via SSL embeddings. (ii) SPEAKER: Realism of speaker identity. (iii) PROSODY: Pitch, duration, and rhythm quality. (iv) INTELLIGIBILITY: Uses ASR-derived features.

---

[*]available at `hf.co/datasets/ttsds/noise-reference`

Each factor is evaluated using multiple feature representations (see Table 1). The scores for these features are averaged to produce a factor score, and these factor scores are in turn averaged to yield the final TTSDS2 score. Thus, TTSDS2 is a *distributional metric* by design, as it compares entire distributions rather than individual samples. Through its use of factor scores, it also functions as a *multi-dimensional perceptual metric*, providing interpretable insights into specific speech attributes, as categorised in Section 1.4.

**Computing Wasserstein distances**    To compare feature distributions, we use the 2-Wasserstein distance ($W_2$), which is intuitively understood as the Earth Mover's Distance (EMD). This aligns with its common application in computer vision, such as the Fréchet Inception Distance (FID) (Heusel et al., 2017). The $W_2$ distance is well-suited for this task due to its desirable properties: it is symmetric (unlike Kullback-Leibler divergence (Kullback, 1951)) and can differentiate between non-overlapping distributions (unlike Jensen-Shannon divergence (Lin, 1991; Kolouri et al., 2019)).

For high-dimensional vectors where distributions are approximated by multivariate Gaussians, the squared 2-Wasserstein distance (also known as the Fréchet distance) between a real dataset $D$ and its synthetic counterpart $\tilde{D}$ is given by:

$$W_2(D, \tilde{D})^2 = \|\mu - \tilde{\mu}\|_2^2 + \mathrm{Tr}\left(\Sigma + \tilde{\Sigma} - 2(\tilde{\Sigma}^{1/2}\Sigma\tilde{\Sigma}^{1/2})^{1/2}\right)$$

where $\mu, \Sigma$ and $\tilde{\mu}, \tilde{\Sigma}$ are the mean and covariance matrices of the real and synthetic feature distributions, respectively. In the one-dimensional case, the squared distance possesses a closed-form solution based on the inverse Cumulative Distribution Functions (CDFs), $C^{-1}$ and $\tilde{C}^{-1}$:

$$W_2(D, \tilde{D})^2 = \int_0^1 (C^{-1}(z) - \tilde{C}^{-1}(z))^2 dz$$

as formulated in (Kolouri et al., 2019).

To normalise distances across different features and factors, we compute a score, ranging from 0 (identical to a noise distribution) to 100 (identical to the real reference distribution). For any synthetic speech feature distribution $\tilde{P}$ – where the lexical content need not match the reference – we compute its $W_2$ distance to each distractor noise dataset in the set $\mathfrak{D}^{\mathrm{NOISE}}$ and denote the minimum as $W_2^{\mathrm{NOISE}} = \min_{D^{\mathrm{NOISE}} \in \mathcal{D}^{\mathrm{NOISE}}} \left[ W_2\left(\tilde{D}, D^{\mathrm{NOISE}}\right)\right]$.

The distance to the real speech distribution $P(\mathcal{R}(S)|D)$ is denoted as $W_2^{\mathrm{REAL}}$. Using these terms, the normalized similarity score for a feature is defined as:

$$\mathrm{TTSDS2}(D, \tilde{D}, \mathfrak{D}^{\mathrm{NOISE}}) = 100 \times \frac{W_2^{\mathrm{NOISE}}}{W_2^{\mathrm{REAL}} + W_2^{\mathrm{NOISE}}} \tag{1}$$

Equation 1 yields scores between 0 and 100, where values above 50 indicate stronger similarity to real speech than to noise. The final TTSDS2 score is the unweighted arithmetic mean of the factor scores. While each factor score requires the computation of five Wasserstein distances, due to the ensembling effect of several representations, only a small number of samples is required, with 50-100 samples being sufficient, which is not the case for distributional scores with a single latent representation Kilgour et al. (2019).

**Updated factors and features**    This work proposes modifications to the original TTSDS framework to make it more robust to several domains for each of its factors. To ensure our metric does not overfit to specific features and to maintain high baseline scores for real data, all feature selections were finalised prior to running any correlation experiments with human judgments. We validated feature robustness by splitting each ground-truth dataset in two and computing the TTSDS score of one half against the other. Any candidate feature that scored below 95 on average, or exhibited a high standard deviation across datasets, was excluded. Following these selection criteria, we updated the feature sets. INTELLIGIBILITY in TTSDS originally relied on Word Error Rate (WER). In preliminary experiments, these WER features resulted in low scores for real data across domains – we use speech recognition models' final-layer activations instead. For PROSODY, TTSDS originally used (a) the WORLD pitch contour (Morise et al., 2016), (b) masked-prosody-model embeddings (Minixhofer

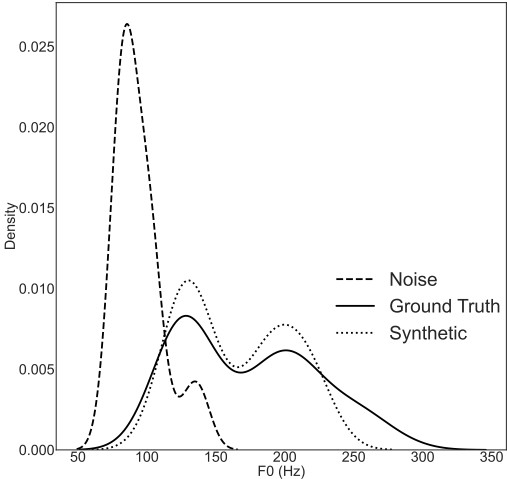

Figure 1: Distribution of $F_0$ in TTSDS for ground-truth, synthetic, and noise datasets.

Table 2: Mean over datasets of MOS, CMOS, SMOS and the corresponding TTSDS2 score.

| System | MOS | CMOS | SMOS | TTSDS2 |
|---|---|---|---|---|
| Ground Truth | 3.70±0.06 | 0.00±0.13 | 4.37±0.15 | 93.21 |
| E2-TTS | **3.41**±0.13 | -0.23±0.18 | 4.37±0.13 | 91.73 |
| Vevo | 3.36±0.14 | **0.08**±0.18 | 4.01±0.15 | 90.20 |
| F5-TTS | 3.33±0.14 | -0.34±0.18 | 4.10±0.15 | 91.16 |
| MaskGCT | 3.28±0.14 | -0.17±0.17 | **4.39**±0.14 | **91.76** |
| FishSpeech | 3.24±0.15 | -0.43±0.21 | 3.58±0.20 | 89.88 |
| TorToiSe | 3.22±0.16 | -0.57±0.25 | 2.73±0.18 | 88.95 |
| VoiceCraft | 3.15±0.15 | -0.44±0.21 | 3.66±0.17 | 88.30 |
| WhisperSpeech | 3.12±0.15 | -0.73±0.27 | 2.68±0.19 | 87.91 |
| HierSpeech++ | 3.08±0.17 | -0.86±0.24 | 3.48±0.21 | 88.63 |
| StyleTTS2 | 3.01±0.15 | -0.66±0.23 | 2.93±0.18 | 85.87 |
| Pheme | 2.99±0.17 | -1.00±0.22 | 3.35±0.18 | 88.84 |
| OpenVoice | 2.92±0.16 | -1.21±0.25 | 2.59±0.19 | 83.32 |
| VALL-E | 2.90±0.14 | -0.60±0.25 | 3.43±0.19 | 83.59 |
| GPTSoVITS | 2.81±0.14 | -0.57±0.21 | 3.82±0.17 | 89.22 |
| XTTS | 2.77±0.17 | -0.73±0.26 | 2.57±0.18 | 88.20 |
| MetaVoice | 2.49±0.14 | -1.23±0.21 | 2.18±0.18 | 87.38 |
| Bark | 2.49±0.14 | -1.12±0.23 | 2.51±0.17 | 85.21 |
| ParlerTTS | 2.39±0.17 | -1.19±0.18 | 2.93±0.17 | 84.88 |
| NaturalSpeech2 | 2.05±0.12 | -1.42±0.21 | 2.06±0.16 | 81.71 |
| SpeechT5 | 1.98±0.15 | -1.56±0.26 | 2.63±0.19 | 84.84 |

et al., 2024), and (c) token lengths (in frames) extracted from HuBERT (Hsu et al., 2021). We found that the token-length features lead to low scores for real speech. We instead compute the utterance-level speaking rate by dividing the number of deduplicated HuBERT tokens in an utterance by the number of frames. We do the same for the multilingual phone recogniser Allosaurus (Li et al., 2020), also included in the original. GENERIC uses the same HuBERT (Hsu et al., 2021) and wav2vec 2.0 (Baevski et al., 2020) features as in the original, but we also add WavLM (Chen et al., 2022) features for increased diversity. The factors and their features are shown in Table 1. For multilingual use, we replace HuBERT with mHuBERT-147 (Boito et al., 2024), and wav2vec 2.0 with its XLSR-53 counterpart (Conneau et al., 2021).

## 3 CORRELATIONS WITH LISTENING TESTS ACROSS DATASETS

We now outline how we validate TTSDS to correlate with human scores across a variety of datasets.

### 3.1 DATASETS FROM READ SPEECH TO CHILDREN'S SPEECH

Since most systems are still trained using audiobook speech, and audiobook speech is easier to synthesize due to its more regular nature (He et al., 2024), we use samples from the LibriTTS (Zen et al., 2019) test split as a baseline. Since LibriTTS is filtered by Signal-to-Noise Ratio (SNR), it only contains clean, read speech. In the remainder of this work, we refer to this as CLEAN. For all datasets, utterances between 3 and 30 seconds with a single speaker are selected. The remaining datasets alter this baseline domain in the following ways:

NOISY is created by scraping LibriVox recordings from 2025 (to avoid their occurrence in the training data) *without* SNR filtering. This tests how evaluation is affected by noise present in the recordings.

WILD is created by scraping recent YouTube videos and extracting utterances, which tests the metrics' ability to generalize to diverse speaking styles and recording conditions. Its data collection and processing are inspired by Emilia (He et al., 2024). We scrape 500 English-language YouTube videos uploaded in 2025 using 10 different search terms which emphasise scripted and conversational speech alike. We perform Whisper Diarization (Ashraf, 2024) to isolate utterances.

KIDS is a subset of the My Science Tutor Corpus (Pradhan et al., 2024) and contains children's conversations with a virtual tutor in an educational setting. This tests if evaluation metrics can generalize to data rarely encountered during the training.

For all systems, we select 100 speakers at random, with two utterances per speaker. We then manually filter the data to exclude content which is (i) difficult to transcribe or (ii) potentially controversial

or offensive. This leaves us with 60 speakers for each dataset. The first utterance by each speaker is used as the reference provided to the TTS system, while the transcript of the second utterance is used as the text to synthesize. This way, we can evaluate both intrusive and non-intrusive metrics. We use matching speaker identities to eliminate any possible preferences of listeners of one speaker over another, and to avoid systems scoring highly merely because of a set of speakers is closer to the reference than for other systems.

## 3.2 Collecting human judgements across systems and datasets

We recruit 200 annotators using Prolific[†] which annotate the ground-truth and synthetic data for 20 TTS systems across the aforementioned datasets, in terms of MOS, CMOS and SMOS. Annotators are screened to be native speakers from the UK or the US and asked to wear headphones in a quiet environment. Any that fail attention checks are excluded. To keep survey duration around 30 minutes, each annotator is assigned to one dataset only, resulting in 50 listeners per dataset. For MOS, there are 6 pages with 5 samples each, one of which is always the ground truth, while the others are selected at random. For CMOS and SMOS, 18 comparisons between ground truth and a randomly selected system's sample are conducted. To avoid any learning or fatigue effects if a certain measure is always asked first or last, the order of the three parts of the test is varied from annotator to annotator. The median completion time was 32 minutes and the annotators were compensated with $10, resulting in an hourly wage of $\approx$ \$19. For both MOS and CMOS, we instruct annotators to rate the *Naturalness* of the speech. MOS and SMOS, in line with recommendations of (Kirkland et al., 2023), are evaluated on a 5-point scale ranging from Bad to Excellent. CMOS is evaluated on a full-point scale ranging from -3 (much worse) to 3 (much better). We collect a total of 11,846 anonymized ratings and utterances, of which we publish 11,282, excluding the ground truth utterances due to licensing. The ratings can be accessed at `hf.co/datasets/ttsds/listening_test`. While we use this data to validate if TTSDS2 aligns with human ratings, future work could use it for improving MOS prediction networks, since, to the best of our knowledge, all publicly available datasets of this size use TTS systems which have not reached human parity (Huang et al., 2024; Tjandra et al., 2025; Cooper et al., 2022; Maniati et al., 2022).

## 3.3 Evaluated objective metrics

We use the VERSA evaluation toolkit (Shi et al., 2024) for all compared objective metrics, except UTMOSv2, which was not included at the time of writing. For *Audiobox Aesthetics* we select their Content Enjoyment (AE-CE), Content Usefulness (AE-CU), and Production Quality (AE-PQ) subscores, which they show to correlate with MOS (Tjandra et al., 2025). For distributional metrics, we evaluate *Fréchet Audio Distance* using Contrastive Language-Audio Pretraining latent representations (Wu et al., 2023). For MOS prediction, we evaluate UTMOS (Saeki et al., 2022), UTMOSv2 (Baba et al., 2024), NISQA (Mittag et al., 2021), DNSMOS (Reddy et al., 2022), and SQUIM MOS (Kumar et al., 2023), which is the only MOS prediction system we evaluate that requires a non-matching reference. For speaker embedding cosine similarity, which require non-matching reference samples as well, we use three systems included in ESPNet-SPK (Jung et al., 2024): RawNet3, ECAPA-TDNN and X-Vectors. We also include some legacy signal-based metrics, which are STOI, PESQ, and MCD – these are the only ones to require matching references. In the next section, we compare these metrics with the subjective evaluation scores.

We evaluate both the original TTSDS metric (Minixhofer et al., 2024) and the updated TTSDS2 version described in Section 2

## 3.4 Correlations

For each TTS of the 20 systems, we average human ratings for MOS, CMOS and SMOS for CLEAN, NOISY, WILD and KIDS. These are the "gold" ratings. We now examine the Spearman correlation coefficients (since we deem ranking the systems most important) of these results with the aforementioned metrics across the four datasets. As Table 3 shows, **TTSDS2** shows the most consistent correlation across the datasets, with an average correlation of 0.67, surpassing the original by 10% relative. All correlations for TTSDS and TTSDS2 are statistically significant with $p < 0.05$.

---

[†]`prolific.com`

Table 3: Spearman rank correlations. Colours: ■ −1 … −0.5, ■ −0.5 … 0, ■ 0 … 0.5, ■ 0.5 … 1.

| Metric | Clean | | | Noisy | | | Wild | | | Kids | | |
|---|---|---|---|---|---|---|---|---|---|---|---|---|
| | MOS | CMOS | SMOS | MOS | CMOS | SMOS | MOS | CMOS | SMOS | MOS | CMOS | SMOS |
| TTSDS2 (Ours) | **0.75** | **0.69** | **0.73** | 0.59 | 0.54 | 0.71 | 0.75 | 0.71 | 0.75 | 0.61 | 0.50 | 0.70 |
| TTSDS Minixhofer et al. (2024) | 0.60 | 0.62 | 0.52 | 0.49 | 0.61 | 0.66 | 0.67 | 0.57 | 0.67 | 0.70 | 0.52 | 0.60 |
| X-Vector | 0.46 | 0.42 | 0.56 | 0.40 | 0.29 | 0.77 | 0.82 | **0.82** | 0.62 | 0.70 | 0.57 | **0.75** |
| RawNet3 | 0.36 | 0.26 | 0.52 | 0.44 | 0.37 | **0.82** | **0.85** | 0.80 | 0.64 | **0.73** | **0.61** | **0.77** |
| SQUIM | 0.68 | 0.46 | 0.37 | 0.48 | 0.48 | 0.60 | 0.62 | 0.75 | **0.79** | 0.57 | 0.55 | 0.45 |
| ECAPA-TDNN | 0.36 | 0.29 | 0.47 | 0.29 | 0.22 | 0.72 | 0.81 | 0.78 | 0.58 | 0.69 | 0.60 | 0.72 |
| DNSMOS | 0.41 | 0.37 | 0.22 | 0.57 | 0.36 | 0.22 | 0.35 | 0.28 | 0.03 | 0.31 | 0.10 | 0.28 |
| AE-CE | 0.60 | 0.46 | 0.32 | 0.58 | 0.53 | 0.21 | 0.19 | 0.10 | 0.11 | -0.02 | -0.12 | -0.10 |
| AE-CU | 0.49 | 0.37 | 0.30 | **0.60** | **0.58** | 0.13 | 0.35 | 0.24 | 0.22 | -0.09 | -0.21 | -0.13 |
| AE-PQ | 0.49 | 0.33 | 0.21 | 0.55 | 0.48 | 0.04 | 0.21 | 0.16 | 0.12 | 0.03 | -0.08 | -0.05 |
| UTMOSv2 | 0.39 | 0.25 | 0.09 | 0.34 | 0.36 | 0.19 | 0.16 | 0.14 | -0.04 | 0.05 | 0.03 | -0.02 |
| FAD (CLAP) | -0.22 | 0.06 | -0.01 | 0.45 | 0.30 | 0.16 | -0.03 | 0.08 | 0.25 | 0.12 | 0.26 | 0.04 |
| UTMOS | 0.51 | 0.30 | 0.31 | 0.47 | 0.29 | 0.00 | -0.12 | -0.12 | -0.26 | -0.02 | -0.18 | -0.04 |
| STOI | -0.11 | 0.01 | 0.02 | -0.06 | 0.00 | 0.19 | 0.07 | 0.41 | 0.24 | -0.32 | -0.08 | 0.05 |
| PESQ | 0.01 | -0.16 | 0.27 | -0.34 | 0.00 | 0.07 | -0.14 | 0.01 | -0.06 | -0.08 | -0.04 | -0.38 |
| NISQA | 0.05 | 0.00 | 0.06 | 0.05 | -0.21 | -0.53 | -0.32 | -0.33 | -0.64 | -0.29 | -0.27 | -0.46 |
| MCD | -0.46 | -0.37 | -0.27 | -0.45 | -0.58 | -0.74 | -0.33 | -0.45 | -0.51 | -0.31 | -0.13 | -0.38 |
| WER | -0.19 | -0.18 | -0.17 | -0.11 | -0.30 | -0.13 | -0.28 | -0.17 | -0.22 | -0.45 | -0.26 | -0.39 |

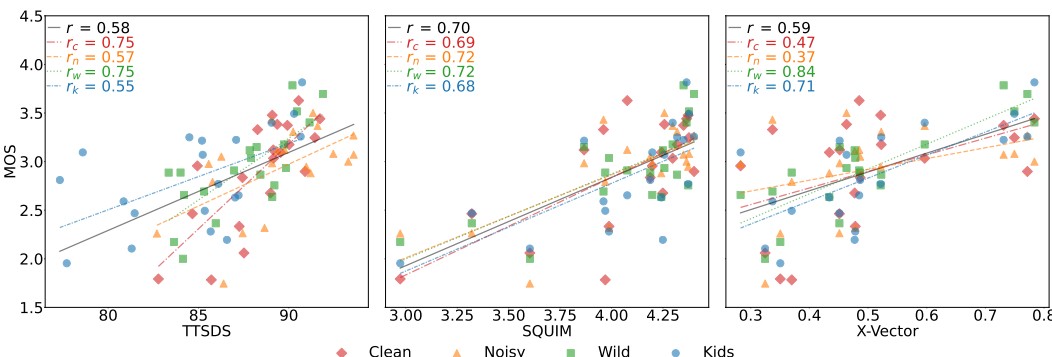

Figure 2: Correlation of three representative objective metrics with human MOS across the four datasets. Each colour/marker denotes a domain. Solid line = overall least-squares fit; dashed/dotted lines = domain-specific fits; each with corresponding Pearson $r$.

**Speaker Similarity** metrics come second, with average correlations of 0.6 for RawNet3 and X-Vector Speaker Similarities. Of the **MOS Prediction networks**, only SQUIM MOS performs well, with an average correlation of 0.57. Following the final Speaker Similarity tested, ECAPA-TDNN, there is a large drop in average correlation, with all remaining averages below 0.3. We note that many of the metrics, including **Audiobox Aesthetics** and **UTMOSv2**, still perform well on NOISY and CLEAN, which only contains audiobook speech. Metrics seem to struggle most on KIDS, which is expected, as it is the furthest removed from the most common TTS domains. When averaging the scores across domains, TTSDS2 agrees with MOS and CMOS for the top 4 and bottom 3 systems, as can be seen in Figure 2.

We also investigate the top-performing TTSDS2, X-Vector, and SQUIM MOS correlations. As Figure 2 shows, some behaviours are not shown in their correlation coefficients alone; TTSDS2 acts the most like a continuous scale; both SQUIM MOS and X-Vector Speaker Similarity show some clustering behaviour. Since both SQUIM and X-Vector are essentially black boxes, we cannot conclusively state what causes this behaviour, but it could indicate overfitting to specific systems.

## 3.5 COMPLEMENTARY FACTORS & GENERALISATION

To validate our design choice of equally weighting all factor scores in the final TTSDS2 calculation, we conduct an ablation study comparing our simple unweighted mean against a learned weighting

Table 4: Leave-One-Out Cross-Validation (Generalisation). Evaluation on a held-out domain after training on the other three.

| Held-Out Domain | Simple Mean (Baseline) | Learned Weights (LOOCV) |
|---|---|---|
| CLEAN | **0.747** | 0.645 |
| NOISY | **0.590** | 0.514 |
| WILD | **0.752** | 0.658 |
| KIDS | 0.606 | **0.853** |

Table 5: Instability of learned weights. Optimal coefficients vary drastically by training domain, occasionally becoming negative.

| Training Domain | Gen. | Speak. | Pros. | Intel. |
|---|---|---|---|---|
| CLEAN | -0.162 | 0.072 | 0.048 | 0.098 |
| NOISY | 0.066 | 0.033 | 0.004 | 0.071 |
| WILD | -0.017 | 0.072 | 0.004 | -0.020 |
| KIDS | 0.050 | 0.101 | -0.032 | 0.026 |
| All Combined | -0.045 | 0.066 | 0.002 | 0.019 |

approach. In the learned setting, factor weights are optimised via linear regression to maximise the correlation with subjective MOS ratings. We evaluate the generalisation capabilities of both aggregation methods using Leave-One-Out Cross-Validation (LOOCV) across our four evaluation domains. As shown in Table 4, the regression model is fit on three domains and evaluated on the fourth held-out domain. The unweighted simple mean outperforms the domain-optimised learned weights in three out of the four unseen domain scenarios, indicating that learned weights tend to overfit to their training distributions. We further examine the stability of these learned weights in Table 5. The optimal coefficients exhibit significant variance depending on the training domain. In several configurations, factors even receive negative coefficients (e.g., the GENERIC factor when trained on CLEAN and WILD), despite these individual factors independently demonstrating positive correlations with human judgements. These results highlight the practical advantage of our ensembling approach. The simple mean effectively acts as a regulariser, mitigating the inherent noise present in individual feature representations across shifting domains. Crucially, this confirms that TTSDS2 can operate as a fully unsupervised metric, requiring no domain-specific fine-tuning or parameter fitting while maintaining strong and stable cross-domain generalisation.

## 4    MULTILINGUAL & RECURRING EVALUATION

While the previous Section outlined robustness across datasets in a single language due to the ease of conducting listening tests in English, we extend TTSDS2 for multilingual use, and provide a public benchmark in 14 languages – this covers all languages synthesised by at least two systems, and is to be extended as more multilingual TTS systems are released.

As our benchmark should be updated frequently to avoid data leakage, and represent a wide range of recording conditions, speakers and environments, we decide to automate the creation of the WILD dataset described in Section 3.1. However, since manual filtering is not feasible in the long term for a growing set of languages and evaluation re-runs, we automate the collection process as can be seen in Algorithm 1, and which we describe in detail in the following section.

Here is the updated pipeline section with the requested pseudo-code and additional details.

### 4.1    PIPELINE

The TTSDS2 pipeline, available at `github.com/ttsds/pipeline`, can be used to rebuild the multilingual dataset regularly. The process is outlined in Algorithm 1 and detailed below.

The pipeline consists of the following automated steps: (i) **Data Scraping:** Ten keywords (see line 2 of Algorithm 1) are translated for each language, and for each term, a YouTube search in the given language, and for a time range starting after the last models' publication in the evaluation set is conducted. The results are sorted by views to avoid low-quality or synthetic results and only videos longer than 20 minutes are included. The beginning and end of each video is trimmed to avoid intro and outro music or artefacts, and are then diarised using Whisper as before. We use FastText (Bojanowski et al., 2016; Grave et al., 2018) language identification on the automatically generated transcripts to filter out videos not in the target language. (ii) **Preprocessing:** We then extract utterances from the middle of the video and only keep utterances from a single speaker as identified in the previous diarisation step. (iii) **Filtering:** Next, the utterances are filtered for potentially offensive or controversial content. We filter the data using XNLI (Conneau et al., 2018; Laurer et al., 2022) with potentially controversial topics as entailment. This leads to a large number of falsely filtered

---

**Algorithm 1** TTSDS2 Benchmark Pipeline

---

**Input:** Languages $L$, TTS Models $M$
1: AllScores ← { }
2: Keywords ← *talk show, interview, debate, sports commentary, news, politics, economy, technology, science, podcast*
3: **for** each language $l$ in $L$ **do**
4:    Keywords ← Translate(Keywords, $l$)
5:    Videos ← SearchYouTube(Keywords, language=$l$, period=lastQuarter, minDuration=20, maxCount=250, sortBy=views)
6:    Utterances ← []
7:    **for** each video in Videos **do**
8:       video ← TrimVideo(start=5, end=5)
9:       DiarizedSegments ← WhisperDiarization(video)
10:      **if** LanguageID(DiarizedSegments) is $l$ **then**
11:         SingleSpeakerSegments ← SelectSingleSpeaker(DiarizedSegments)
12:         Utterances.append(ExtractUtterances(SingleSpeakerSegments, max=16))
13:      **end if**
14:   **end for**
15:   CleanUtterances ← FilterControversial(Utterances, method=XNLI)
16:   CleanUtterances ← FilterCrosstalk(CleanUtterances, method=Pyannote)
17:   CleanUtterances ← FilterMusic(CleanUtterances, method=Demucs)
18:   MatchedPairs ← SelectSpeakerPairs(CleanUtterances, count=50)
19:   ReferenceSet, SynthesisSet ← SplitPairs(MatchedPairs)
20:   **for** each model $m$ in $M$ **do**
21:      SyntheticAudio ← Synthesise(m, ReferenceSet, SynthesisSet)
22:      Score ← CalculateTTSDS2(SyntheticAudio, ReferenceSet)
23:      AllScores[$m, l$] ← Score
24:   **end for**
25: **end for**
26: **Return:** AllScores

---

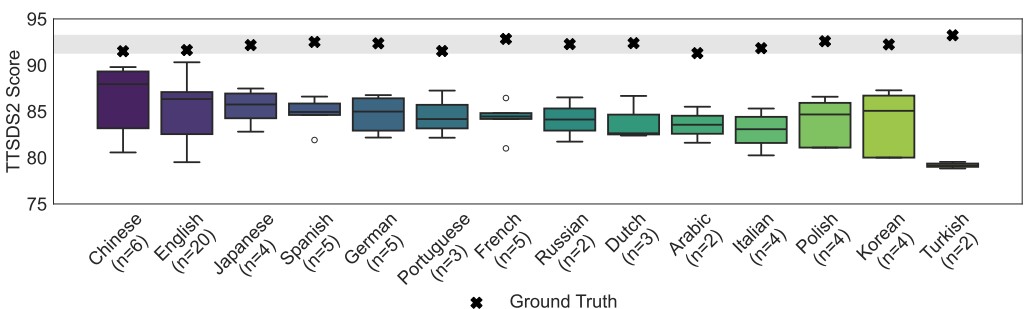

Figure 3: TTSDS2 scores across 14 languages. $n$ indicates the number of systems per language.

texts, which in our case is not a downside, since we only want a small number of "clean" samples. Finally, we use Pyannote speaker diarisation (Bredin, 2023) to detect if there is any crosstalk, and Demucs (Rouard et al., 2023) source separation, to detect if there is any background music. Of the remaining utterances, 50 speaker-matched pairs are selected for each language, and split into the REFERENCE and SYNTHESIS set. (iv) **Synthesis:** For all systems in the benchmark, we synthesise the speaker identities in REFERENCE with the text in SYNTHESIS. (v) **TTSDS2:** We apply multilingual TTSDS2 to arrive at scores for each, and publish the results at `ttsdsbenchmark.com`. This can be repeated regularly with systems published prior to the data range of the collected video data, to eliminate the possibility of data contamination. We plan to expand to more systems and languages each evaluation round.

## 4.2 Multilingual validity of TTSDS2

While collecting gold MOS labels for 14 languages is out of scope for this work, we verify TTSDS2's applicability to the multilingual case using Uriel+ (Khan et al., 2024), which supplies *typological distances* for the languages evaluated. We show that if TTSDS2 distances correlate to language distances found by linguists, finding that when comparing the ground truth language datasets to each other using TTSDS2, the scores correlate with the distances with $\rho = -0.39$ for regular and $\rho = -0.51$ (both significant with $p < 0.05$) for multilingual TTSDS2 discussed in Section 2 – the negative correlations are expected since a higher score correlates with a smaller distance, and the higher correlation of multilingual TTSDS2 scores is encouraging. Additionally, ground-truth scores are within a narrow range across languages, and scores decrease for lower-resource languages as expected, which is shown in Figure 3.

## 5 Limitations

Since TTSDS2 extracts several features for each utterance and uses CPU-bound Wasserstein distance computations, it uses more compute than other methods. Future work could explore more compute-efficient methods such Maximum Mean Discrepancy (MMD) (Gretton et al., 2006) which has shown promising results in computer vision evaluation (Jayasumana et al., 2024). While it robustly correlates with human evaluation, it never surpasses Spearman correlation coefficient of 0.8, indicating there is either a component of listening tests that is inherently noisy, or not predicted by any objective metric as of yet, and therefore TTSDS2 is not equivalent to, nor can it replace, subjective evaluation. Additionally, some modern TTS systems can have failure cases which are not identifiable as such by TTSDS2 such as when the transcript given is not reproduced faithfully. To mitigate this shortcoming, we report the number of utterances with high Word Error Rates for each system at ttsdsbenchmark.com. TTSDS2 currently also does not capture the context the utterances were spoken in, and does not include long-form samples, as many systems do not support generation of utterances beyond 30 seconds.

## 6 Ethics Statement

To mitigate the risk of generating novel, unattributed speech, the text synthesized for a given speaker's voice is always sourced from a separate, distinct utterance previously spoken by that same individual. This ensures no new statements are created. Furthermore, we do not redistribute any of the original ground-truth audio samples, to respect the terms of their original release. We also acknowledge that advancements in TTS technology carry dual-use potential. However, the TTSDS2 metric is designed to evaluate distributions of speech (i.e., entire datasets) rather than individual samples. This characteristic makes it poorly suited for the iterative development of single deepfake utterances. Conversely, its distributional nature may offer utility in identifying large-scale synthetic speech campaigns, serving as a potential detection tool. Finally, the selection of the 14 languages was based on the availability of multiple open-source TTS systems. We recognize that this may inadvertently reflect existing biases in speech technology research. By providing an open-source and extensible pipeline, we aim to empower the community to apply and expand this benchmark to a wider, more inclusive set of languages.

## 7 Conclusion

In this work, we introduce TTSDS2, a robust, factorised metric that demonstrates consistently high correlation with human judgments across a wide variety of speech domains. Our extensive evaluation shows that out of 16 state-of-the-art objective metrics, TTSDS2 is the only one to maintain a strong Spearman correlation ($\rho > 0.5$) in every tested condition, achieving an average of $\rho \approx 0.67$. This consistency holds true not only for clean audiobook speech but also for challenging domains including noisy, conversational, and children's speech, where many existing metrics fail to achieve significant correlations. To further advance reproducible and inclusive research, we extend the framework for multilingual use and provide a public, recurring benchmark covering 14 languages. This benchmark is supported by an automated pipeline that can continually renew the dataset to prevent contamination and ensure long-term relevance. By providing a reliable objective measure that aligns closely with human perception, we hope the TTSDS2 benchmark and its associated resources will enhance the efficiency and direction of future research in human-quality text-to-speech synthesis.

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

# A EVALUATED TTS SYSTEMS AND PARITY WITH REAL SPEECH

Table 6: Open-source TTS systems, prior evaluation, and results for each system relative to ground-truth (GT) speech: † = accompanied by publication; ∗ = third-party implementation; Parity column: Reported MOS/CMOS are close to GT ($\sim$), surpassing GT ($>$) or below GT ($<$)

| System | Year | Objective | Subjective | Parity |
|---|---|---|---|---|
| Bark[Suno (2023); Mylo (2023)] | 2023 | | | |
| †∗E2-TTS[Eskimez et al. (2024); Chen et al. (2024)] | 2024 | ✓ | ✓ | $\sim$ |
| †F5-TTS[Chen et al. (2024)] | 2024 | ✓ | ✓ | $>$ |
| FishSpeech 1.5[Liao et al. (2024)] | 2024 | | ✓ | $<$ |
| GPT-SoVITS v2[RVC-Boss (2024)] | 2024 | | | |
| †HierSpeech++ 1.1[Lee et al. (2023)] | 2023 | ✓ | ✓ | $\sim$ |
| †MaskGCT[Wang et al. (2024); Zhang et al. (2024)] | 2024 | ✓ | ✓ | $>$ |
| MetaVoice-1B[Sharma et al. (2024)] | 2024 | | | |
| †∗NaturalSpeech 2[Shen et al. (2024); Zhang et al. (2024)] | 2023 | ✓ | ✓ | $\sim$ |
| OpenVoice v2[Qin et al. (2023)] | 2024 | | | |
| †∗ParlerTTS Large 1.0[Lacombe et al. (2024); Lyth & King (2024)] | 2024 | ✓ | ✓ | $>$ |
| †Pheme[Budzianowski et al. (2024)] | 2024 | ✓ | | |
| †SpeechT5[Ao et al. (2022)] | 2022 | | ✓ | $<$ |
| †StyleTTS 2[Li et al. (2023)] | 2023 | ✓ | ✓ | $>$ |
| TorToiSe[Betker (2023)] | 2022 | | | |
| †∗VALL-E[Wang et al. (2023a); Zhang et al. (2024)] | 2024 | ✓ | ✓ | $<$ |
| †Vevo[Zhang et al. (2025)] | 2024 | ✓ | ✓ | $<$ |
| †VoiceCraft-830M[Peng et al. (2024)] | 2024 | ✓ | ✓ | $<$ |
| WhisperSpeech Medium[Cłapa et al. (2024)] | 2024 | | | |
| †XTTS-v1[Casanova et al. (2024)] | 2023 | ✓ | | |

The systems evaluated in TTSDS2 are shown in Table 6, subject to expansion. It should be noted that for ParlerTTS, a modification of the codebase was used to allow for voice cloning, which could degrade performance. The latest available checkpoint before `01-01-2025`, the data collection cut-off, was used for each system, with the exception of XTTSv2, which we experienced difficulties with for grapheme-to-phoneme conversion.

Of the 20 systems, 13 were accompanied by research papers, of which all but 3 reported subjective and objective evaluation. Of the 13 systems accompanied by papers, 7 report being within 0.05 of ground truth MOS or CMOS (Eskimez et al., 2024; Shen et al., 2024; Lee et al., 2023) or surpassing ground truth by this margin (Chen et al., 2024; Wang et al., 2024; Li et al., 2023; Lyth & King, 2024). Of the objective evaluation methods outlined in Section 1.4, *WER* and *Speaker Similarity* were reported 5 times, followed by *UTMOS*, *CER*, *Fréchet*-type distances, and *MCD*, which were all reported twice. *PESQ* and *STOI* were reported once.

# B  LISTENING TEST DETAILS

For the precise wording of questions, an example survey can be viewed at `ttsdsbenchmark.com/survey`. For WILD "audio books" was replaced with "YouTube"; for KIDS with "children's speech". We also include instructions here:

Listing 1: MOS Instructions

```
You will be listening to 6 sets of audio samples. In each set, there are
    5 audio recordings of the same text. Some of them may be synthesized
    while others may be from audio books. Please listen to the audio
    clips carefully, then,

Rate how natural each audio clip sounds on a scale of 1 to 5 with 1
    indicating completely unnatural speech (bad) and 5 completely natural
     speech (excellent). Here, naturalness includes whether you feel the
    speech is spoken by a real speaker from a human source.

Some clips may be cut early, please ignore words that may be cut off when
     rating the naturalness.

There are some audios for checking your attention through the survey. If
    a certain number of the scores are rated unreasonably, you will not
    be paid. Please listen to the audios carefully and do not fill out
    the survey randomly.
```

Listing 2: CMOS Instructions

```
You will be listening to 18 sets of audio samples. In each set, there are
     two audio recordings (A and B) of the same text. Some of them may be
     synthesized while others may be from audio books. Please listen to
    the audio clips carefully, compare each audio with the reference, and
     then

Rate how natural is A compared to B on a scale of -3 to 3 with 3
    indicating that A is much better than B. Here, naturalness includes
    whether you feel the speech is spoken by a real speaker from a human
    source, as opposed to being synthesized by a computer.

Some clips may be cut early, please ignore words that may be cut off when
     rating the naturalness.

There are some audios for checking your attention through the survey. If
    a certain number of the scores are rated unreasonably, you will not
    be paid. Please listen to the audios carefully and do not fill out
    the survey randomly.
```

Listing 3: SMOS Instructions

```
You will be listening to 18 sets of audio samples. In each set, there are
     two audio recordings (A and B) of the same text. Some of them may be
     synthesized while others may be from audio books. Please listen to
    the audio clips carefully, compare each audio with the reference, and
     then

Rate how similar the speaker in B is to the speaker in A on a scale of 1
    to 5 with 5 indicating that the speaker in B is the same person as
    the speaker in A. Here, similarity includes whether you feel the
    speech is spoken by the same person.

Some clips may be cut early, please ignore words that may be cut off when
     rating the similarity.

There are some audios for checking your attention through the survey. If
    a certain number of the scores are rated unreasonably, you will not
    be paid. Please listen to the audios carefully and do not fill out
    the survey randomly.
```

### B.1 MOS, CMOS AND SMOS

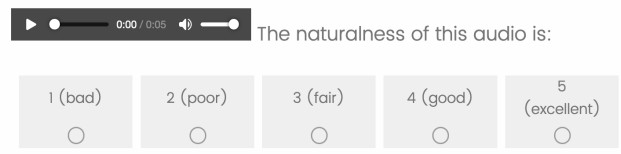

Figure 4: Interface for Mean Opinion Score (MOS) listening tests.

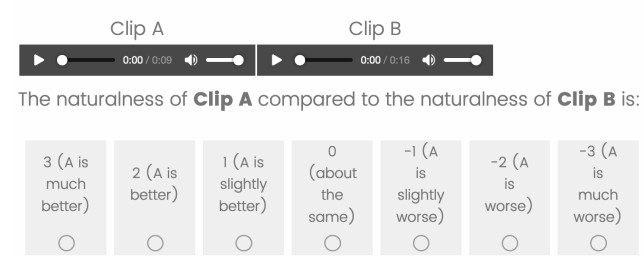

Figure 5: Interface for Comparison MOS (CMOS) listening tests.

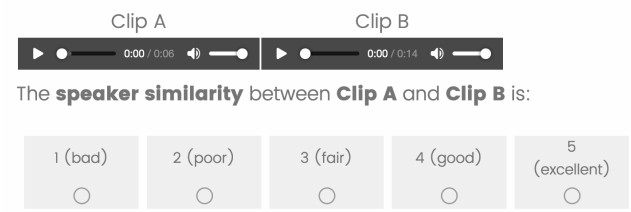

Figure 6: Interface for Speaker Similarity MOS (SMOS) listening tests.

In this work, we conduct listening test using the most common methodologies for subjective speech evaluation, namely the Mean Opinion Score (MOS), Comparison MOS (CMOS), and Speaker Similarity MOS (SMOS). For a MOS test, listeners rate isolated audio samples on a 5-point scale from 1 (bad) to 5 (excellent), as shown in Figure 4. CMOS tests present listeners with two samples, A and B, and ask them to rate their relative naturalness on a 7-point scale from -3 (A is much worse than B) to +3 (A is much better than B), which can be seen in Figure 5. This is particularly useful for fine-grained comparisons when absolute scores may saturate. SMOS is used for voice cloning evaluation and operates similarly to CMOS, but listeners rate the speaker similarity between two clips on a 5-point scale, as illustrated in Figure 6.

### B.2 ETHICS APPROVAL

This study was certified according to the Informatics Research Ethics Process, reference number 112246. Participants were informed about the purpose and terms of the study using the Participant Information Sheet available here: `ttsdsbenchmark.com/PIS.pdf`. None of the participants withdrew their consent within the 30-day period following the listening test study.

## C   DOMAIN-WISE MOS, CMOS, SMOS AND TTSDS2 RESULTS

Table 7: Listening test results in terms of MOS, CMOS and SMOS relative to ground truth speech.

| System | Clean | | | Noisy | | | Wild | | | Kids | | |
|---|---|---|---|---|---|---|---|---|---|---|---|---|
| | MOS | CMOS | SMOS | MOS | CMOS | SMOS | MOS | CMOS | SMOS | MOS | CMOS | SMOS |
| Ground Truth | 3.72±0.06 | 0.00±0.11 | 4.29±0.14 | 3.35±0.06 | -0.10±0.21 | 4.45±0.19 | 3.75±0.06 | -0.08±0.12 | 4.51±0.11 | 3.96±0.06 | 0.18±0.09 | 4.24±0.16 |
| E2-TTS | 3.44±0.12 | -0.02±0.18 | **4.45±0.11** | 3.00±0.14 | -0.21±0.18 | 4.29±0.13 | 3.40±0.12 | -0.26±0.17 | **4.53±0.11** | **3.82±0.13** | -0.42±0.20 | 4.19±0.19 |
| Vevo | 3.37±0.12 | 0.10±0.18 | 4.05±0.12 | 3.07±0.17 | -0.06±0.18 | 4.45±0.13 | **3.78±0.12** | **0.30±0.17** | 3.65±0.19 | 3.22±0.16 | **-0.03±0.17** | 3.91±0.17 |
| F5-TTS | 3.25±0.14 | -0.17±0.13 | 4.36±0.16 | 3.08±0.16 | 0.07±0.18 | 4.02±0.15 | 3.52±0.12 | -0.30±0.20 | 4.42±0.13 | 3.49±0.14 | -0.95±0.23 | 3.60±0.19 |
| MaskGCT | 2.90±0.16 | -0.36±0.17 | 4.43±0.13 | 3.27±0.13 | 0.19±0.16 | **4.50±0.12** | 3.69±0.13 | -0.16±0.17 | 4.30±0.13 | 3.26±0.14 | -0.37±0.20 | **4.33±0.18** |
| FishSpeech | 3.03±0.14 | -0.35±0.20 | 3.53±0.19 | 3.37±0.17 | -0.48±0.24 | 3.85±0.19 | 3.18±0.13 | -0.14±0.20 | 3.49±0.21 | 3.40±0.16 | -0.76±0.21 | 3.46±0.19 |
| TorToiSe | 3.38±0.15 | **0.30±0.21** | 3.30±0.19 | **3.50±0.16** | 0.02±0.25 | 2.79±0.19 | 2.93±0.15 | -1.44±0.33 | 2.65±0.18 | 3.07±0.17 | -1.17±0.22 | 2.17±0.16 |
| VoiceCraft | 3.18±0.13 | -0.68±0.20 | 3.69±0.15 | 3.31±0.11 | -0.36±0.16 | 3.33±0.17 | 2.87±0.17 | -0.24±0.17 | 3.67±0.17 | 3.25±0.17 | -0.47±0.28 | 3.95±0.18 |
| WhisperSpeech | 3.12±0.16 | -1.07±0.22 | 3.07±0.18 | 2.98±0.15 | -0.22±0.27 | 2.90±0.16 | 3.15±0.16 | -0.98±0.27 | 2.57±0.21 | 3.22±0.12 | -0.67±0.31 | 2.20±0.19 |
| HierSpeech++ | **3.63±0.18** | -0.91±0.20 | 4.04±0.21 | 3.12±0.14 | -0.29±0.26 | 3.71±0.18 | 2.91±0.18 | -0.63±0.26 | 2.71±0.22 | 2.65±0.17 | -1.64±0.23 | 3.46±0.23 |
| StyleTTS2 | 3.33±0.12 | -0.81±0.19 | 4.05±0.15 | 3.43±0.15 | **0.38±0.23** | 3.12±0.19 | 2.69±0.15 | -1.05±0.26 | 2.66±0.21 | 2.59±0.16 | -1.17±0.25 | 1.90±0.16 |
| Pheme | 3.48±0.16 | -0.24±0.17 | 3.31±0.21 | 2.95±0.12 | -0.93±0.23 | 3.88±0.14 | 2.76±0.22 | -0.88±0.20 | 3.59±0.18 | 2.77±0.19 | -1.94±0.30 | 2.63±0.21 |
| OpenVoice | 2.96±0.14 | -1.40±0.20 | 3.63±0.18 | 2.98±0.19 | -0.19±0.31 | 2.54±0.22 | 2.66±0.13 | -1.60±0.23 | 2.00±0.19 | 3.09±0.15 | -1.63±0.24 | 2.17±0.18 |
| VALL-E | 2.84±0.13 | -0.23±0.19 | 4.14±0.15 | 3.05±0.15 | -0.64±0.21 | 3.50±0.19 | 2.89±0.13 | -0.77±0.27 | 3.34±0.25 | 2.81±0.16 | -0.76±0.34 | 2.76±0.18 |
| GPTSoVITS | 3.09±0.14 | 0.00±0.16 | 3.67±0.19 | 2.88±0.13 | -0.33±0.21 | 4.24±0.14 | 2.64±0.11 | -0.93±0.22 | 3.98±0.17 | 2.63±0.15 | -1.02±0.25 | 3.39±0.17 |
| XTTS | 2.68±0.15 | -0.46±0.27 | 2.52±0.16 | 3.11±0.15 | -0.76±0.27 | 2.61±0.18 | 3.12±0.19 | -0.37±0.21 | 2.48±0.17 | 2.20±0.19 | -1.34±0.26 | 2.68±0.20 |
| MetaVoice | 2.33±0.13 | -1.71±0.15 | 2.14±0.16 | 2.32±0.15 | -1.07±0.26 | 2.53±0.19 | 3.04±0.13 | -0.75±0.21 | 2.06±0.21 | 2.28±0.15 | -1.37±0.24 | 2.00±0.15 |
| Bark | 1.78±0.12 | -1.10±0.20 | 2.61±0.18 | 2.79±0.13 | -0.74±0.23 | 2.45±0.18 | 2.89±0.14 | -1.09±0.24 | 3.10±0.17 | 2.49±0.14 | -1.56±0.25 | 1.88±0.16 |
| ParlerTTS | 2.46±0.18 | -1.02±0.15 | 3.10±0.18 | 2.26±0.12 | -0.84±0.18 | 3.62±0.17 | 2.37±0.19 | -1.25±0.20 | 2.40±0.17 | 2.47±0.19 | -1.65±0.20 | 2.59±0.18 |
| NaturalSpeech2 | 1.79±0.11 | -1.35±0.24 | 2.14±0.14 | 2.26±0.17 | -0.76±0.24 | 2.41±0.17 | 2.17±0.11 | -1.82±0.17 | 1.77±0.13 | 1.95±0.11 | -1.75±0.21 | 1.90±0.20 |
| SpeechT5 | 2.06±0.14 | -1.24±0.24 | 2.79±0.16 | 1.75±0.15 | -1.91±0.27 | 2.61±0.18 | 2.00±0.16 | -1.69±0.24 | 1.79±0.14 | 2.11±0.15 | -1.39±0.28 | 3.33±0.27 |

The Table above shows the raw MOS, CMOS and SMOS scores derived from the listening tests.

# D   TTSDS2 FACTORS

Table 8: Pearson correlation ($r$) between each factor and MOS.

| Dataset | Generic | Speaker | Prosody | Intelligibility |
|---------|---------|---------|---------|-----------------|
| Clean | 0.42 | **0.84** | 0.38 | 0.47 |
| Noisy | 0.59 | **0.86** | 0.46 | 0.59 |
| Wild | 0.53 | **0.59** | 0.34 | 0.58 |
| Kids | **0.80** | 0.70 | 0.60 | 0.63 |

To assess their impact on the overall score, we present their individual TTSDS2 factors' Pearson $r$ correlations in Table 8. For CLEAN and NOISY, the speaker factor dominates – an interesting result, given that paired Speaker Similarity metrics performed poorly for these datasets. For WILD and KIDS, the speaker factor is less useful, with Intelligibility and Generic showing similar correlations. Prosody is highest for KIDS, suggesting its value in assessing replication of children's prosodic patterns. Overall, while the Speaker factor is generally the most helpful, the others are complementary, especially for the non-read speech in WILD and KIDS. All correlations are significant ($p < 0.05$).

# E   CORRELATIONS BETWEEN OBJECTIVE AND SUBJECTIVE METRICS

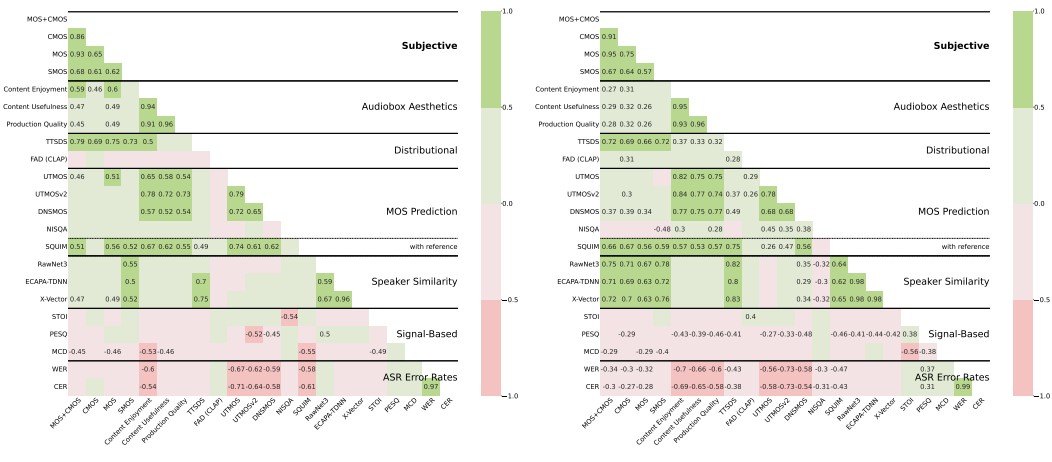

(a) In-domain (Clean) correlations.      (b) Cross-domain (Noisy, Wild, Kids) correlations.

Figure 7: Significant ($p < 0.05$) Spearman correlation heatmaps between metrics.

As shown in Figure 7, in the in-domain setting (CLEAN), TTSDS2 achieves the highest correlation with subjective scores ($\rho = 0.79$), outperforming both MOS and CMOS in predicting each other. Audiobox Aesthetics also correlates with MOS, though less strongly than previously reported, likely due to differences in evaluation datasets. Among MOS prediction networks, only SQUIM performs well ($\rho = 0.66$), while X-Vector is the only Speaker Similarity metric showing moderate correlation. In the out-of-domain datasets (NOISY, WILD, KIDS), correlation patterns shift: Audiobox Aesthetics weakens, while SQUIM improves ($\rho = 0.67$). Speaker Similarity metrics – especially RawNet3 – show the strongest correlations ($\rho = 0.73$). TTSDS2 remains strongly correlated ($\rho = 0.72$), but is no longer the top predictor.

## E.1   NEGATIVE CORRELATIONS WITH MCD AND WER

For Word Error Rates (WER) a negative correlation is expected, as higher WER indicates worse TTS performance. However, the correlations observed (see Table 3) are still not sufficient to consistently predict subjective ratings, with a minimum of $\rho = -0.45$ for KIDS MOS. Signal-based metrics (MCD, PESQ) occasionally show negative correlations as well, which could be due to "oversmoothing", with systems predicting the average of many possible utterances achieving a better score in said metrics, while leading to worse human ratings.

## F COMPUTE

**Synthesising** samples for all TTS systems in this work, across datasets and languages, including failed runs and reruns, used 28.8 hours on a single A100 GPU. We estimate each rerun to use approximately 8 compute hours for synthesis, subject to a changing number of languages and TTS systems.

**Running TTSDS2** is CPU-bound due to the Wasserstein distance computations. Each TTSDS2 score computation took $\approx$ 9.4 minutes using an `Intel(R) Xeon(R) CPU E5-2620 v4 @ 2.10GHz`.

## G LANGUAGE DISTANCE VISUALISATION

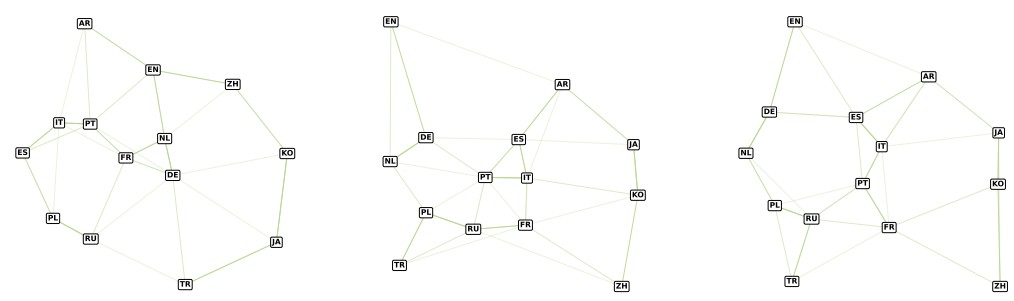

Figure 8: Multidimensional scaling (MDS) distance plots between languages (left to right) for i) Uriel+ typological distances ii) TTSDS2 without multilingual changes iii) multilingual TTSDS2. The three closest neighbors of each language are connected via lines.

As discussed in Section 4, TTSDS2 scores, when interpreted as distances between ground truth language datasets, correlate with Uriel+ typological distances. Figure 8 shows these distances visually. An interesting effect is that English is more distant from other languages than in Uriel+, likely due to the much larger amount of English data used even when training explicitly multilingual models.

## H FILTERING OF CONTROVERSIAL CONTENT

In preliminary experiments, we find LLM-based filtering to be too resource-intensive for a recurring benchmark, while toxic comment classification work does fails to filter potentially controversial but not explicitly toxic content, and often is not available in multiple languages. As detailed in Section 4.1, we therefore use an entailment model. The entailment words used are as follows `negative,political,gender,religion,sexual,controversial,rare word,incomplete,race` The threshholds used are `[0.6, 0.6, 0.6, 0.6, 0.6, 0.8, 0.6, 0.9, 0.6]` and are set using spot-checking on English data. The multilingual entailment model used is `hf.co/MoritzLaurer/mDeBERTa-v3-base-mnli-xnli`.

