# OpenReview forum: "TTSDS2: Resources and Benchmark for Evaluating Human-Quality Text to Speech Systems"
_ICLR.cc/2026/Conference — ICLR 2026 Oral_

### Official Review · Reviewer_KrsR · 2025-10-31

**Soundness:** 3
**Presentation:** 1
**Contribution:** 2
**Rating:** 2
**Confidence:** 3

**Summary:**

This paper introduces TTSDS2, a comprehensive distributional metric for evaluating text-to-speech (TTS) quality. TTSDS2 integrates four key quality dimensions—generic quality, speaker similarity, prosody, and intelligibility—and, for each dimension, leverages multiple embedding models to compute the Wasserstein distance between embeddings of synthesized and reference speech. To assess the robustness of the metric, the authors compile a diverse evaluation dataset encompassing read speech, noisy speech, YouTube speech, and children’s speech, and benchmark 20 TTS systems using both human (MOS) and automatic metrics. Experimental results demonstrate that TTSDS2 consistently correlates strongly with human judgments across domains. The authors also provide a continually updatable evaluation pipeline designed to mitigate data leakage and maintain a reliable benchmark over time.

**Strengths:**

- **Comprehensive and Diverse Evaluation:**

    The paper introduces a test set with human annotations covering four diverse domains—read speech, noisy speech, YouTube speech, and children’s speech. Both the dataset and the evaluation are publicly released, enabling future reproducibility and comparison. The study also benchmarks a wide range of baseline TTS models and compares against multiple existing quality metrics, ensuring a thorough evaluation.

- **Strong Correlation with Human Judgments:**

    TTSDS2 demonstrates consistently high correlation with human perceptual ratings across different speech domains, confirming its robustness and reliability as an automatic TTS quality metric.

- **Continually Updatable Benchmark:**

    The proposed benchmark pipeline supports continual updates while preventing data leakage, which makes it a valuable and sustainable resource for future TTS research and model evaluation.

**Weaknesses:**

- **Poor Writing Quality and Structure:**

    While the main contribution of the paper is clear, the overall writing is repetitive and poorly structured, particularly in Section 1.

- **Questionable Gaussian Assumption in Metric Design:**

    TTSDS2 assumes that the speech embeddings follow a multivariate Gaussian distribution when computing the Wasserstein distance. However, this assumption may not hold in practice, and the authors provide no empirical justification for it. A more principled alternative would be to employ an optimal transport algorithm to estimate the Wasserstein distance directly from the embedding distributions.

- **Unconvincing Multilingual Evaluation:**

    The paper’s claim of multilingual validity for TTSDS2 is not sufficiently supported. To substantiate the multilingual generalization of the proposed metric, additional human evaluations across languages are necessary.

- **Potential Annotation Bias:**

    Each annotator is assigned to only one dataset, which could introduce domain-specific bias and limit cross-domain consistency in the human evaluation results.

**Questions:**

- **Motivation for Distributional Metric:**

    What is the underlying intuition or theoretical justification for why a *distributional* metric should better reflect human perception compared to point-wise measures?

- **Averaging Strategy Across Dimensions:**

    The paper averages scores across different embedding models within each dimension, and then across the four dimensions. Have the authors explored using weighted combinations or learned weights to better capture the relative importance of each dimension?

- **Speaker Similarity Performance:**

    The speaker similarity metrics consistently rank as the second-best in correlation with human judgments. Can the authors provide an explanation or analysis for why this particular dimension performs slightly worse than the others?

---

> ### Author Response · Authors · 2025-11-22
>
> > While the main contribution of the paper is clear, the overall writing is repetitive and poorly structured, particularly in Section 1.
>
> We would be grateful for any feedback that could help us improve this section.
>
> > TTSDS2 assumes that the speech embeddings follow a multivariate Gaussian distribution when computing the Wasserstein distance. However, this assumption may not hold in practice, and the authors provide no empirical justification for it. A more principled alternative would be to employ an optimal transport algorithm to estimate the Wasserstein distance directly from the embedding distributions.
>
> When employing latent features and Wasserstein distance, it is a common assumption used in previous works that latent neural network representations follow a multivariate Gaussian distribution -- see [Heusel et al., 2018](https://arxiv.org/pdf/1706.08500) for a detailed explanation. We build on this work, as using an optimal transport algorithm is quite computationally expensive with high-dimensional samples, and it would not have been possible comparing the same number of systems and domains using such an algorithm.
>
> > The paper’s claim of multilingual validity for TTSDS2 is not sufficiently supported. To substantiate the multilingual generalization of the proposed metric, additional human evaluations across languages are necessary.
>
> We agree that validation with subjective scores is needed in the future, however, we see the correlation with Uriel+ typological distances (see Appendix G) as a promising sign for multilingual applicability.
>
> > Each annotator is assigned to only one dataset, which could introduce domain-specific bias and limit cross-domain consistency in the human evaluation results.
>
> We believed assigned annotators to one dataset each was the most principled approach, as otherwise, it would be possible that TTSDS2 only correlates for ratings for *one specific set* of annotators, rather than four distinct ones.
>
> > What is the underlying intuition or theoretical justification for why a distributional metric should better reflect human perception compared to point-wise measures?
>
> Any models ability to capture the underlying distribution of the data should result in more realistic outputs and therefore higher subjective ratings. We are not aware of a theoretical justification of point-wise measures, since speech generation is a one-to-many problem, and there is no one correct waveform to compare against.
>
> > The paper averages scores across different embedding models within each dimension, and then across the four dimensions. Have the authors explored using weighted combinations or learned weights to better capture the relative importance of each dimension?
>
> This question was raised by several reviewers, so we compared the proposed unweighted **Simple Mean** against a **Learned Weight** approach (optimizing factor weights via linear regression to maximize MOS correlation).
>
> **Table 1: Leave-One-Out Cross-Validation (Generalization)**
> We trained weights on three datasets and evaluated performance on the fourth held-out domain. The unweighted Simple Mean outperforms the learned weights in 3 out of 4 domains.
>
> | Held-Out Domain | Simple Mean (Baseline) $\rho$ | Learned Weights (LOOCV) $\rho$ | $\Delta$ |
> | :--- | :---: | :---: | :---: |
> | **Clean** | **0.747** | 0.645 | $-0.10$ |
> | **Noisy** | **0.590** | 0.514 | $-0.08$ |
> | **Wild** | **0.752** | 0.658 | $-0.09$ |
> | **Kids** | 0.606 | **0.853** | $+0.25$ |
>
> **Table 2: Instability of Learned Weights**
> The optimal weights vary drastically depending on the training domain, with factors occasionally receiving negative coefficients (e.g., *Generic* in Clean/Wild, although on its own, it correlates with MOS).
>
> | Training Domain | Generic | Speaker | Prosody | Intelligibility |
> | :--- | :---: | :---: | :---: | :---: |
> | **Clean** | -0.162 | 0.072 | 0.048 | 0.098 |
> | **Noisy** | 0.066 | 0.033 | 0.004 | 0.071 |
> | **Wild** | -0.017 | 0.072 | 0.004 | -0.020 |
> | **Kids** | 0.050 | 0.101 | -0.032 | 0.026 |
> | **All Combined** | -0.045 | 0.066 | 0.002 | 0.019 |
>
> This shows that unherent noise which is naturally present in the indiviual scores/representations is balanced out by using a simple mean rather than any learned combination. Additionally, this means TTSDS2 is fully unsupervised and does not necessarily require fine-tuning at any stage.
>
> > The speaker similarity metrics consistently rank as the second-best in correlation with human judgments. Can the authors provide an explanation or analysis for why this particular dimension performs slightly worse than the others?
>
> We noticed the speaker similarity performing *better* than most other objective metrics, and slightly worse than TTSDS2. To investigate this further we visualised the point-wise correlations of speaker embeddings and TTSDS2 in figure 2 and lines 360-365.
>
> We would like to thank you for your review and look forward to any further feedback or suggestions.

---

> > ### Comment · Reviewer_KrsR · 2025-11-25
> >
> > Thank you for the response with additional details. I have one more question
> >
> > > We believed assigned annotators to one dataset each was the most principled approach
> > >
> >
> > Why not randomly distribute the instances in one dataset to all annotators to get rid of annotator bias?

---

> > > ### Author Response · Authors · 2025-11-25
> > >
> > > In that case, we would still need to limit the amount of samples such that the survey can be completed in ~30 minutes. This combined with 4 domains with 20 system each means some annotators might not be presented with the same set (or same proportions of) TTS systems, which could lead to other issues. I.e. imagine one annotator randomly gets mostly samples from the top 5 systems, and another mostly samples from the bottom 5. Simply keeping the proportions equal throughout wouldn’t work due to the large number of domain x system combinations and the 30 minute time limit.

---

> > > > ### Comment · Reviewer_KrsR · 2025-11-26
> > > >
> > > > Got it. Thank you. I've raised the score given your response.

---

> > > > > ### Author Response · Authors · 2025-11-26
> > > > >
> > > > > Thank you very much. Please feel free to provide further feedback if anything else comes to mind.

---

### Official Review · Reviewer_LwgN · 2025-11-01

**Soundness:** 3
**Presentation:** 4
**Contribution:** 3
**Rating:** 8
**Confidence:** 3

**Summary:**

This paper introduces TTSDS2, a factorized, distributional objective metric for evaluating modern TTS systems. It measures similarity to real speech across multiple perceptual dimensions (generic, speaker, prosody, intelligibility) using Wasserstein distance over model-derived embeddings. The authors evaluate 20 recent open-weight TTS models across four domains and show that TTSDS2 is the only metric achieving Spearman > 0.5 consistently against MOS/CMOS/SMOS. The paper also provides a multilingual benchmark (14 languages) and an automated pipeline for recurring evaluation. Overall, I believe the work is timely, well-executed, and useful.

**Strengths:**

* The problem is well-motivated: subjective evaluation is expensive and rapidly becoming insufficient as systems approach human quality. The factorized distributional approach is intuitive and gives interpretable sub-scores.

* The empirical evaluation is broad, covering 20 recent models and multiple domains (clean, noisy, wild, children).

* Demonstrating consistent correlation >0.5 with human ratings across all domains is compelling.

* Benchmark and pipeline release are valuable to the community and likely to be widely used.

* Writing is clear and situates the work well within the literature.

**Weaknesses:**

* A more principled justification or ablation is warranted as the choice of feature set can appears somewhat tuned for correlation.
* Figure 3 is not referenced anywhere in text and the caption is vague.
* The multilingual evaluation lacks validation with human preference tests (e.g., CMOS / MUSHRA), making it difficult to verify whether TTSDS2 maintains reliability beyond English.

**Questions:**

Are there clear failure cases where TTSDS2 disagrees with MOS/CMOS? If so, what characteristics do these samples/systems share?

---

> ### Author Response · Authors · 2025-11-22
>
> > A more principled justification or ablation is warranted as the choice of feature set can appears somewhat tuned for correlation.
>
> In lines 206-215, we give an (admittedly brief) overview over the factor selection. To adress your concern, we can provide some additional information as well: To check robustness before any correlations with subjective evaluation, we compared the TTSDS scores of each factor with the score achieved on **real data**. I.e. we split each real dataset in two and compare the score between them. Factors that scored below 95 on average or showed high standard deviation between datasets were excluded. Factor selection was completely finalised before running any correlation experiments. Due to the high number of possible feature sets we believe this is one of the few ways we can ensure there is no overfitting to specific features while also maintaining (desireable) high scores for real data.
>
> > Figure 3 is not referenced anywhere in text and the caption is vague.
>
> Thank you for pointing this out. The figure shows a box plot of TTSDS2 scores for each language. N is the number of systems for each language. We will add these details to the text/caption.
>
> > The multilingual evaluation lacks validation with human preference tests (e.g., CMOS / MUSHRA), making it difficult to verify whether TTSDS2 maintains reliability beyond English.
>
> We agree that validation with subjective scores is needed in the future, however, we see the correlation with Uriel+ typological distances (see Appendix G) as a promising sign for multilingual applicability.
>
> > Are there clear failure cases where TTSDS2 disagrees with MOS/CMOS? If so, what characteristics do these samples/systems share?
>
> There are no clear failure cases we discovered, however, correlations are notably lower for datasets with more environmental noise (Noisy & Kids). This result could be either due to subjective ratings penalising noisy outputs even when they are "natural", or due to some of the feature sets encoding environmental/noise conditions even when another factor is targeted (e.g. speaker embeddings are well known to encode recording/environmental conditions, not just speaker identity).
>
> We would like to thank you for your review and look forward to any further feedback or suggestions.

---

### Official Review · Reviewer_jomi · 2025-11-06

**Soundness:** 3
**Presentation:** 3
**Contribution:** 3
**Rating:** 6
**Confidence:** 3

**Summary:**

This paper introduces TTSDS2, a framework for evaluating text-to-speech (TTS) systems. While TTS has advanced rapidly, existing evaluation metrics often fail to capture the human-level quality of recent systems (particularly in multilingual contexts). Moreover, subjective evaluations such as MOS are difficult to compare across systems due to differing evaluators and datasets. TTSDS2 addresses these issues by assessing how closely the distributions of various perceptual factors (e.g., speaker identity, prosody, intelligibility) in synthetic speech resemble those in real speech, relative to noise. Experimental results demonstrate that TTSDS2 correlates strongly with human listening tests and consistently outperforms other open-source objective metrics.

**Strengths:**

- Comprehensive comparison against numerous evaluation metrics.
- Evaluation across 20 recent TTS systems and 14 languages.
- Diverse and well-structured test sets covering clean, noisy, wild, and children’s speech domains.
- Offers a promising, scalable solution for objective TTS evaluation.
- Shows clear and consistent alignment with subjective (human) evaluation results.

**Weaknesses:**

- The overall methodology closely resembles the original TTSDS, with limited conceptual novelty.
- The paper averages multiple perceptual factor scores into a single TTSDS2 score but provides no justification for using equal weighting.
- The selection criteria for the feature sets within each factor are not clearly explained or motivated.

**Questions:**

- Why do the authors use *noise* samples as the negative anchor? Wouldn’t failed or unnatural TTS outputs serve as a more realistic negative baseline?
- How does the inference or computation time of TTSDS2 compare to other objective evaluation metrics?
- What is the rationale behind averaging all factor scores equally to form the final TTSDS2 score? Are all perceptual dimensions equally important for perceived quality?
- Could the authors report the correlation of each individual factor (e.g., prosody, intelligibility) with mean opinion scores to clarify which aspects contribute most to perceptual alignment?

---

> ### Author Response · Authors · 2025-11-22
>
> > The selection criteria for the feature sets within each factor are not clearly explained or motivated.
>
> In lines 206-215, we give a brief overview over the factor selection. To check robustness before any correlations with subjective evaluation, we compared the TTSDS scores of each factor with the score achieved on **real data**. I.e. we split each real dataset in two and compare the score between them. Factors that scored below 95 on average or showed high standard deviation between datasets were excluded. Factor selection was completely finalised before running any correlation experiments.
>
> > Why do the authors use noise samples as the negative anchor? Wouldn’t failed or unnatural TTS outputs serve as a more realistic negative baseline?
>
> The advantage of noise samples is that they are agnostic to the model and domain in use. Failed or unnatural TTS outputs inherently bias the results based on which TTS model(s) might be used, which language the outputs are in, and potential other confounding factors.
>
> > How does the inference or computation time of TTSDS2 compare to other objective evaluation metrics?
>
> Appendix F shows an example inference time of 10 minutes using an ``Intel(R) Xeon(R) CPU E5-2620 v4 @
> 2.10GHz`` to evaluate each TTS system - this is slower than most objective evaluation metrics, although other distributional metrics can also be computationally expensive. We believe this is still a good trade-off as meaningful evaluation metrics can help save resources on the much more computationally expensive tasks of training models. Additionally, TTSDS2 is CPU-bound, which means it could be run while training a model using GPU resources.
>
> > What is the rationale behind averaging all factor scores equally to form the final TTSDS2 score? Are all perceptual dimensions equally important for perceived quality?
>
> This question was raised by several reviewers, so we compared the proposed unweighted **Simple Mean** against a **Learned Weight** approach (optimizing factor weights via linear regression to maximize MOS correlation).
>
> **Table 1: Leave-One-Out Cross-Validation (Generalization)**
> We trained weights on three datasets and evaluated performance on the fourth held-out domain. The unweighted Simple Mean outperforms the learned weights in 3 out of 4 domains.
>
> | Held-Out Domain | Simple Mean (Baseline) $\rho$ | Learned Weights (LOOCV) $\rho$ | $\Delta$ |
> | :--- | :---: | :---: | :---: |
> | **Clean** | **0.747** | 0.645 | $-0.10$ |
> | **Noisy** | **0.590** | 0.514 | $-0.08$ |
> | **Wild** | **0.752** | 0.658 | $-0.09$ |
> | **Kids** | 0.606 | **0.853** | $+0.25$ |
>
> **Table 2: Instability of Learned Weights**
> The optimal weights vary drastically depending on the training domain, with factors occasionally receiving negative coefficients (e.g., *Generic* in Clean/Wild, although on its own, it correlates with MOS).
>
> | Training Domain | Generic | Speaker | Prosody | Intelligibility |
> | :--- | :---: | :---: | :---: | :---: |
> | **Clean** | -0.162 | 0.072 | 0.048 | 0.098 |
> | **Noisy** | 0.066 | 0.033 | 0.004 | 0.071 |
> | **Wild** | -0.017 | 0.072 | 0.004 | -0.020 |
> | **Kids** | 0.050 | 0.101 | -0.032 | 0.026 |
> | **All Combined** | -0.045 | 0.066 | 0.002 | 0.019 |
>
> This shows that the ensembling approach we use has merit, as some inherent noise which is naturally present in the indiviual scores/representations is balanced out by using a simple mean rather than any learned combination. Additionally, this means TTSDS2 is fully unsupervised and does not necessarily require fine-tuning at any stage.
>
> > Could the authors report the correlation of each individual factor (e.g., prosody, intelligibility) with mean opinion scores to clarify which aspects contribute most to perceptual alignment?
>
> We believe we answered this in Appendix D, which we repeat here:
>
> **Table 3: Pearson correlation ($r$) between each factor and MOS.**
>
> | Dataset | Generic | Speaker | Prosody | Intelligibility |
> | :--- | :---: | :---: | :---: | :---: |
> | Clean | 0.42 | **0.84** | 0.38 | 0.47 |
> | Noisy | 0.59 | **0.86** | 0.46 | 0.59 |
> | Wild | 0.53 | **0.59** | 0.34 | 0.58 |
> | Kids | **0.80** | 0.70 | 0.60 | 0.63 |
>
> To assess their impact on the overall score, we present their individual TTSDS2 factors’ Pearson $r$ correlations in Table 3. For CLEAN and NOISY, the speaker factor dominates – an interesting result, given that paired Speaker Similarity metrics performed poorly for these datasets. For WILD and KIDS, the speaker factor is less useful, with Intelligibility and Generic showing similar correlations. Prosody is highest for KIDS, suggesting its value in assessing replication of children’s prosodic patterns. Overall, while the Speaker factor is generally the most helpful, the others are complementary, especially for the non-read speech in WILD and KIDS. All correlations are significant ($p < 0.05$).
>
> We would like to thank you for your review and look forward to any further feedback or suggestions.

---

> > ### Comment · Reviewer_jomi · 2025-11-27
> >
> > Thank you to the authors for the detailed explanations. I have a few additional comments:
> >
> > - Could the authors clarify why multiple features are selected within each factor rather than choosing a single representative feature?
> >
> > - I remain unconvinced about using noise as the negative anchor for evaluating the overall quality of synthesized speech. It seems that alternative negative anchors—such as degraded speech produced through signal processing—might be more appropriate for this framework. What are the authors’ thoughts on this?
> >
> > - Appendix F reports the inference speed of the proposed system, but does not provide comparable measurements for other systems. Including inference speed for the baselines would make the evaluation more complete and informative.

---

> > > ### Author Response · Authors · 2025-12-03
> > >
> > > > Could the authors clarify why multiple features are selected within each factor rather than choosing a single representative feature?
> > >
> > > Due to a low number of samples (to match them with listening tests) -- inherent noise which is naturally present in the indiviual scores/representations is balanced out by using multiple representative features.
> > >
> > > > I remain unconvinced about using noise as the negative anchor for evaluating the overall quality of synthesized speech. It seems that alternative negative anchors—such as degraded speech produced through signal processing—might be more appropriate for this framework. What are the authors’ thoughts on this?
> > >
> > > We believe measuring the latent space behaviour in an unopinionated way is only possible using noise, all zeros or all ones, as we do. Otherwise new domains, languages, etc. would always require new distractors, when our fully algorithmically generated samples work in practice.
> > >
> > > > Appendix F reports the inference speed of the proposed system, but does not provide comparable measurements for other systems. Including inference speed for the baselines would make the evaluation more complete and informative.
> > >
> > > We have not seen extensive performance comparison of different evaluation metrics in the literature, as they are usually much less computationally expensive than training, so we don't think this addition is vital, although it could be an interesting exploration for future work.

---

### Author Response · Authors · 2025-12-03

We would like to thank the reviewers for their constructive feedback and are pleased with the consensus that TTSDS2 offers a robust, scalable solution for objective TTS evaluation. We are encouraged that the reviewers recognised the comprehensive nature of our benchmark and the clear, consistent alignment with subjective human results across diverse domains.

During the discussion, we addressed the key questions regarding our methodology:

  -  Weighting Strategy: We demonstrated via Leave-One-Out Cross-Validation that our unweighted Simple Mean outperforms learned weights. This confirms that the inherent noise naturally present in individual representations is balanced out more effectively by a simple mean, ensuring the metric remains fully unsupervised.

  -  Feature Selection: We clarified that factor selection was completely finalised before running any correlation experiments to ensure there was no overfitting to specific features.

Following these clarifications, Reviewer KrsR raised their score, further solidifying the positive consensus. We believe TTSDS2 offers the community a robust, scalable, and interpretable standard for modern TTS evaluation.

---

### Meta-Review · Area_Chair_EHyB · 2026-01-08

**Summary:**

This paper receives 3 high-quality reviews, with 2 positive initial ratings (8, 6) and 1 negative initial rating (2).

The 1 negative reviewer has participated in the rebuttal discussions actively. According to the reviewer's responses, this reviewer's concerns have been addressed and the reviewer is willing to raise the rating.

The rest of reviewers initially have positive ratings, and the authors have provided good responses regarding to the reviewers' concerns.

**Reviewer Concerns:**

All reviewers' concerns have been addressed well.

**Reviewer Scores:**

Two reviewers give positive initial ratings, and the negative reviewer's concerns have been addressed.

---

### Decision · Program_Chairs · 2026-01-26

Accept (Oral)